# Uncovering bridging diseases in complex multimorbidity pathways: A network science approach

Javier Alvarez-Galvez[1,2]*, Javier Arroyo[3,4]

1 Computational Social Science DataLab (CS2 DataLab), University Institute for Sustainable Social Development, University of Cádiz, Cádiz, Spain, 2 Department of General Economy (Sociology), Faculty of Nursing and Physiotherapy, University of Cádiz, Cádiz, Spain, 3 Department of Computer Science, University of Alcalá, Madrid, Spain, 4 Institute of Knowledge Technology, Complutense University of Madrid, Madrid, Spain

* javier.alvarezgalvez@uca.es

## Abstract

Multimorbidity, the co-occurrence of multiple chronic diseases, represents a significant challenge in healthcare, necessitating advanced analytical methods for a better understanding. Although numerous studies focus on characterizing chronicity profiles across different population groups, there is still a need to identify specific diseases that play a crucial role in shaping multimorbidity patterns. This study applies network science to analyze multimorbidity structures and identify bridging diseases that facilitate the development of complex multimorbidity patterns, using data from a representative sample of 2,200 individuals aged 50 and older residing in southern Spain. Our findings reveal significant gender-based differences in multimorbidity patterns, with women experiencing a higher burden of chronic diseases, resulting in more complex and tightly linked disease networks. The analysis highlights the relevance of specific conditions, such as liver dysfunction in men and depression in women, as key contributors to the formation of complex multimorbidity structures. These findings emphasize the importance of sex/gender-specific healthcare strategies aimed at controlling and preventing diseases that may act as catalysts for multisystem multimorbidity, which have a profound impact on both mortality rates and healthcare utilization.

## Introduction

The study of multimorbidity, understood in a broad sense as the presence of two or more chronic conditions in an individual, has become a priority in recent years [1]. Multimorbidity is linked to the increase in life expectancy and the progressive aging of populations [2,3] and is currently one of the challenges of current health systems due to the complexity of the management of pluripathological patients [4]. As recent studies have shown, this condition is associated with worse prognoses of physical

**Data availability statement:** Public use of health data used in the study has not been permitted by the Andalusian Biomedical Research Ethics Coordinating Committee (Comité Coordinador de Ética de la Investigación Biomédica de Andalucía) (PEIBA:2249-N-19)

since this dataset contain potentially identifying information (sex, data of birth, postal code, health diagnosis); however, if necessary, a new request for the data could be submitted to the institutional email proyectos.investigacion@uca.es for subsequent submission and approval by the corresponding committee

**Funding:** The author(s) received no specific funding for this work.

**Competing interests:** The authors have declared that no competing interests exist.

and mental health, disability, poorer quality of life, greater frequentation in the use of health services, polymedication, and fragmentation in health care [5–6], and therefore presents great economic and human costs for our social and health systems [7–9].

In this context, in recent years there has been increasing interest in the identification and characterization of multimorbidity patterns [4,10–16]. Although traditionally the pathways of certain patterns such as cardiovascular or musculoskeletal have been well identified by traditional techniques such as factor analysis and cluster analysis [10], the incorporation of more sophisticated classification techniques such as latent class analysis has enabled a better characterization of the so-called complex multimorbidity patterns (also called multisystem because they combine different patterns) [4,17,18]. In addition, the incorporation of new classification techniques from the new field of machine learning such as Markov chains [19], association rule models [20], or the use of mixed graphical models [21] are enabling progress in the characterization of these complex patterns which, on the one hand, are more unpredictable for expert healthcare personnel and, on the other, consequently more lethal for the patient [4]. Indeed, complex patterns generate a greater overload for health services, since they make self-management of care more difficult and end up overloading the health system itself and reducing the quality of life of patients [7].

This problem highlights the need to identify the pathways to complex multimorbidity. Central to these pathways are 'bridging diseases,' which are conditions that connect and integrate various multimorbidity patterns, playing a crucial role in linking and transforming these patterns. Starting from the hypothesis of the existence of these 'bridging diseases,', this study is based on a cross-sectional descriptive design, whose objective is: (1) to characterize through the implementation of mixed graphical models (MGM) the latent structure of interrelationships between co-occurring chronic diseases that configure the multimorbidity patterns of the population of the province of Cadiz; (2) to conduct a gender-based analysis to identify specific patterns, acknowledging the pre-existing differences between genders, and (3) to identify through the use of network centrality metrics the bridging diseases that enable the transition to complex (or multisystem) patterns of multimorbidity.

## Methods

### Study design

This study is based on a descriptive cross-sectional design. For this work, we used data from the DEMMOCAD survey, a geographically representative study based on computer-assisted telephone interviews (CATI). The target population was adults and older individuals living in the province of Cadiz, Southern Spain. The province of Cadiz, located in southern Spain, has a population of 1,254,866 inhabitants (as of 2024).

### Sample size and selection criteria

Considering the variability of municipalities in this area, we developed a stratified sampling strategy that included the six administrative regions of the province. For this population, we estimated a minimum sample size of 1600 people for a

confidence level of 95% and an estimation error of ± 2.5 units. This threshold was chosen to ensure higher precision in identifying multimorbidity patterns and balancing statistical reliability with sample representativeness. This minimum sample was extended by an additional 600 individuals to increase the number of people having multiple chronic conditions in the sample while reaching quotas in the different geographical areas. In total, 2200 people aged 50 and older were finally interviewed between February 2 and February 27, 2022.

Considering that our study focuses on the multimorbidity population, one of the selection criteria required participants to be over 50 years old to ensure a minimum level of chronicity burden in this population group. From the initial sample of 2,200 respondents, we selected for this study those with two or more chronic diseases to consider exclusively the population with multimorbidity. After excluding participants without multimorbidity, the final sample consisted of 1592 respondents. This sample size is necessary to guarantee the representativeness of the survey for the whole of the geographical area analysed.

This survey data was collected as a part of the DEMMOCAD project (ITI-0028–2019), which aimed to characterise multimorbidity patterns and their social inequalities in this geographical area. This research was authorised by the Andalusian Biomedical Research Ethics Coordinating Committee (Comité Coordinador de Ética de la Investigación Biomédica de Andalucía) (PEIBA:2249-N-19). After the ethic committee approval, we obtained verbal consent from participants via phone call, recording their willingness to participate. Non-participation was voluntary, and participants were informed of their right to withdraw at any time. Their response was recorded via phone call and data were anonymized to prevent identification of the participants. The response rate for the telephone survey was 13.3%, reflecting the proportion of completed interviews relative to all eligible individuals contacted.

## Variables of interest

For the study of patterns of multimorbidity, a total of 32 variables referring to diagnosed chronic diseases were selected: 1) Hypertension; 2) Infarction; 3) Coronary disease; 4) Other heart; 5) Varicose veins; 6) Arthrosis; 7) Cervical pain; 8) Lumbar pain; 9) Allergy; 10) Asthma; 11) Respiratory disease; 12) Diabetes; 13) Stomach ulcer; 14) Urinary incontinence; 15) Cholesterol; 16) Cataracts; 17) Skin condition; 18) Constipation; 19) Hepatic dysfunction; 20) Depression; 21) Anxiety; 22) Other mental; 23) Stroke; 24) Migraine; 25) Haemorrhoids; 26) Cancer; 27) Osteoporosis; 28) Thyroid disease; 29) Kidney disease; 30) Prostate/Menopausal problems (for men or women respectively); 31) Accidents; 32) Obesity. The indicators were binary and were therefore measured as 0 'Has no disease' or 1 'Has the disease' In addition, we also considered the gender dimension in order to identify possible inequalities in the multimorbidity patterns of men and women. Specifically, these diseases were selected in order to compare the results with those obtained in previous studies in Spain [4,7,10].

## Statistical analysis

Although there are different techniques to analyse the latent structure of interrelationships between the different chronic conditions (e.g., clustering techniques, latent class analysis, and factor models, among others) [4,10–20], our study required a method that not only grouped diseases but also visualised their associations. Thus, following the previous work of Alvarez-Galvez and Vegas Lozano [21], we chose to use mixed graphical models to estimate the *k-order* Mixed Graphical Model (MGM) of multimorbidity patterns based on disease pairwise associations.

This graphical modelling framework explicitly estimates conditional dependencies between diseases while controlling for confounding effects. From the variables in the analysis, the *mgm* package [22] creates network models via a nodewise estimation method that incorporates a penalty based on least absolute shrinkage and selection operator regularization (LASSO). To improve interpretability and reduce spurious correlations, the *glasso* algorithm (or graphical lasso) is used to create sparser and less complex networks. This method forces low partial correlation coefficients to zero, enhancing network sparsity and controlling graph density through fitting parameters. This approach improves interpretability by reducing false associations and capturing the most meaningful disease relationships. Furthermore, regarding conditional

independence, MGMs rely on a Markov random field structure to represent conditional relationships between variables. Formally, the model ensures that a variable is conditionally independent of the rest of the variables given its neighbourhood in the network.

The resulting MGM consists of nodes (chronic conditions) and edges (relationships between variables) [23]. The width of the edges represents the strength of the association between the variables (i.e., the nodes), so the absence of connections implies that there are no relationships or that they are too weak to be significant. Specifically, edges capture partial correlations in these models, i.e., the correlation between pairs of variables when controlling for all other possible associations in the dataset which, compared to conventional correlation analysis, reduces the risk of spurious correlations. In addition, to enhance the robustness of our results, we incorporated cross-validation to ensure that our findings remain consistent even under small changes in the data.

Once the models are calculated, networks showing pairwise associations between chronic conditions are represented using the *qgraph* package in R [24]. After the relational structure of the disease network for men and women was obtained, we identified specific multimorbidity patterns using a community identification algorithm (cluster_fast_greedy in R) that detects subgraphs or communities within the network that represent prevalent multimorbidity patterns. This approach allowed us to obtain subsets of adjacent diseases, enabling the identification of coherent disease neighbourhoods–that have been identified in specialised literature–while simultaneously providing visual information on how various diseases clustered together [21]. As observed in previous works, this visual approach to the classification of multimorbidity patterns has the advantage of being able to obtain classifications of diseases that are more closely related but, at the same time, without losing sight of other associations that may be relevant in the configuration of other more complex (multisystem) patterns and of the possible role of diseases that may act as bridges between the various disease communities [21]. In this way, we can obtain classifications that are more transparent and that reveal the real complexity of the different disease patterns.

Given the challenge of interpreting disease relevance in multimorbidity patterns, we included an extended centrality analysis that would allow us to identify the most relevant conditions in the resulting structure of multimorbidity patterns. Specifically, the following centrality metrics were incorporated: (1) Degree: the number of direct connections a node has (i.e., number of connection with other diseases); (2) Closeness centrality: the inverse of the average shortest path distance from a node to all other nodes in the network (which means that diseases with high closeness centrality can quickly interact with all other disease in the network); (3) Betweenness centrality: the number of times a node acts as a bridge along the shortest path between two other nodes (i.e., diseases with high betweenness centrality are critical in the combination of multimorbidity patterns, as they control many pathways between disease communities); (4) PageRank: a measure to identify authorities in the network (i.e., disease with high PageRank are considered more important, as they are linked to by many other important nodes); (5) Eigenvector centrality: an indicator of the influence of a node in a network, where connections to high-scoring nodes contribute more to the score of the node (i.e., this metric indicates the role of the disease in the overall connectivity with and influence over other diseases within the network); and (6) Hub score: a measure that identifies nodes (i.e., diseases) that link to many important nodes (hubs) and nodes that are linked from many important nodes (authorities). This analysis was performed through the *igraph* library [25].

Finally, we employed logistic regression to assess the effect of bridging diseases in the subsequent configuration of complex multimorbidity patterns (i.e., multisystem multimorbidity characterised by the combination of two or more multimorbidity patterns). In other words, through the logistic regression model, we aimed to quantify the contribution of variables with high betweenness centrality to the explanation of the complex multimorbidity pattern in men and women. To this end, we also incorporated socio-economic and demographic control variables, such as age, education, and income, allowing us to provide a more comprehensive characterisation of the social profiles analysed.

## Results

### Disease prevalence

Fig 1 illustrates differences in disease prevalence between males and females. Overall, women over 50 years of age experience a greater disease burden than men of the same age in all recorded conditions, except for heart attacks. Specifically, the most prevalent conditions for men are hypertension, cholesterol, and low back pain, followed closely by cervical pain, osteoarthritis, and diabetes. In the case of women, the most prevalent conditions are the same, although with the difference that all these chronic diseases present prevalences above 20% (something that in men only occurs in hypertension). Therefore, the visual comparison underscores significant gender differences in the prevalence of specific physical and mental health conditions (e.g., migraine, anxiety, depression, thyroid disease, varicose veins, allergies, osteoporosis, and arthrosis, among others).

### Detecting multimorbidity patterns

Fig 2 shows the disease association networks obtained from the mixed graphical models for men and women. Associations between chronic diseases in men demonstrate lower overall connectivity in the chronic disease network (with

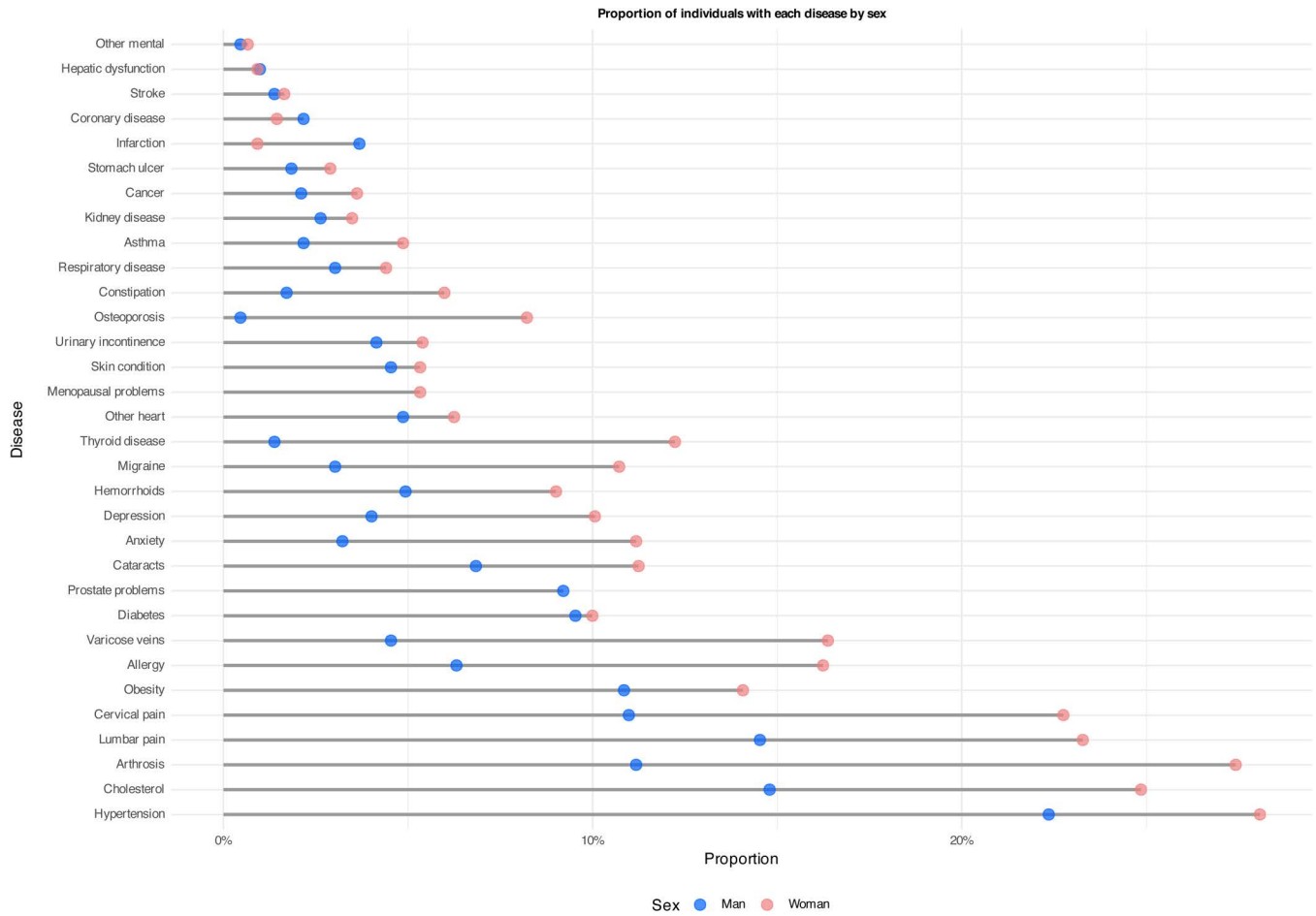

**Fig 1. Differences in disease prevalence by gender.**

seven conditions fully disconnected from others). Although in the resulting graph certain connectivity is observed between diseases that respond to the same multimorbidity pattern, for example, cardiovascular (heart attack, stroke, other heart, coronary heart disease, and diabetes), musculoskeletal (osteoarthritis, low back pain, and neck pain), respiratory (allergy, asthma and respiratory disease) or mental (anxiety, depression and other mental), a certain transversality is also observed in conditions such as hypertension (D1) or hepatic dysfunction (D19) which, despite being respectively linked to heart and digestive related conditions, also has multiple comorbid relationships with other diseases that fall outside the cardiovascular or digestive patterns. This cross-cutting connectivity suggests that men with these transversal primary conditions are more likely to suffer from multiple additional health issues and subsequently complex multimorbidity. The dense clustering around these core diseases points to a high degree of interconnectivity between different health conditions, which places them in a key position to act as possible bridges to the configuration of complex multimorbidity patterns.

On the right side, the chronic disease network of women shows greater interconnectivity than that of men, which has to do with the higher prevalence of all conditions in the group of women. In general, a better definition of disease profiles linked to a single multimorbidity pattern is observed. For example, it is clear that cardiovascular, musculoskeletal, respiratory, and mental illnesses are relatively well grouped at the local level. This is less common in the male disease network where only a number of diseases end up clustered together. Moreover, while diseases such as hypertension and diabetes also appear as significant nodes in the graph, health conditions like osteoporosis, migraine, mental and menopausal problems also emerge as possible bridging diseases for women. These findings suggest, on the one hand, that women experience broader and more complex multimorbidity patterns than men, while, on the other hand, they exhibit patterns with a distinctive character in relation to both mental and musculoskeletal health conditions. These differences in the patterns of men and women highlight the existence of important gender-specific health inequalities.

To assess the reliability of the MGM, we evaluated its classification accuracy in modelling co-occurring chronic diseases. The resulting models demonstrated satisfactory classification results, with accuracy values ranging from 0.64 to 0.99 for the man sample. Women's disease network showed similar results, with accuracy values ranging from 0.62 to

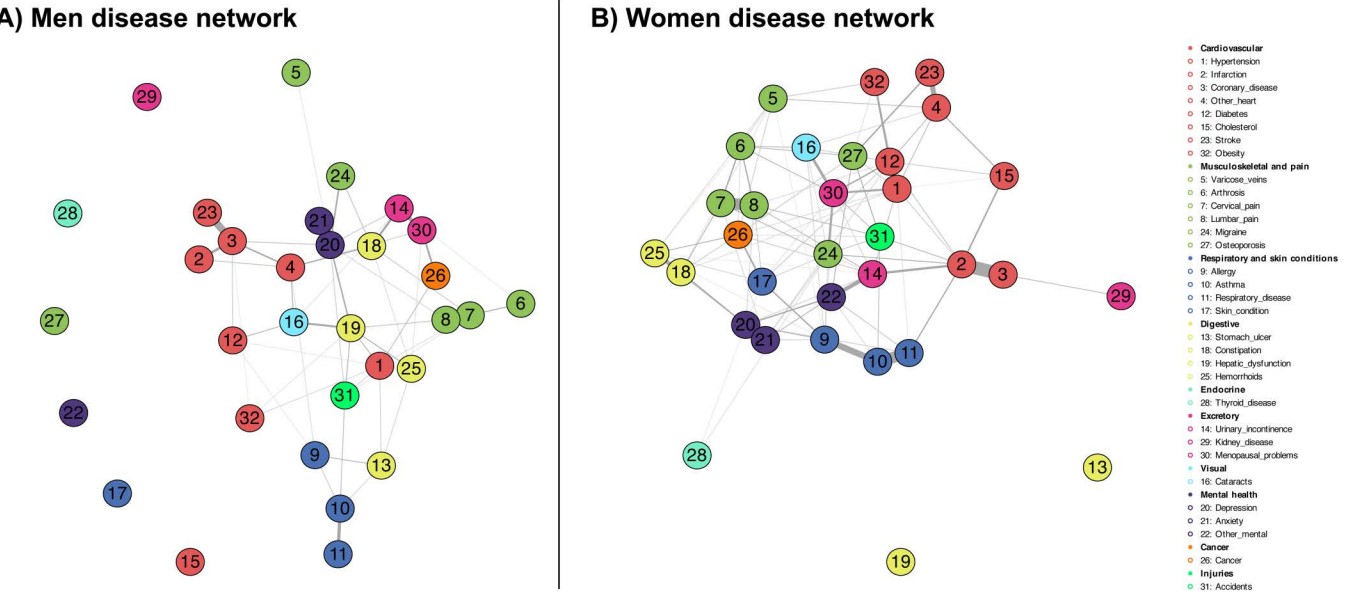

**Fig 2. Disease association obtained from the mixed graphical models for men and women.**

0.98. Overall, the high accuracy values indicate the general robustness of the MGM, especially considering the significant variability of the predictors included in the same analysis.

In Fig 3, disease communities–represented by different colours–reveal distinct patterns of multimorbidity by sex. In men, we observe well-defined clusters such as Cardiovascular, Cardiometabolic complex (which combines cardiometabolic and other conditions such as hepatic dysfunction and cataracts), Respiratory, Musculoskeletal, and Mental. Notably, cancer forms a distinct cluster with a significant overlap with other conditions (prostate problems or constipation). The interconnectedness within the Cardiovascular and Cardiometabolic disease communities indicates a strong association between these health conditions in men. Conversely, the disease communities are more interlaced in women, with the Cardiovascular and Cardiometabolic conditions forming a large, complex cluster that includes diseases like hypertension, diabetes, and obesity, which may indicate a higher prevalence and interconnection between these conditions. The Musculoskeletal disease neighbour is also extensive in women, integrating conditions such as arthritis and osteoporosis, while showing significant overlap with the cardiometabolic and mental diseases. Furthermore, the Mental health cluster is tightly connected with the Respiratory and Cardiometabolic clusters, suggesting a broader interplay between mental health and physical health conditions among women. Similarly, the Cancer pattern forms a more integrated cluster, overlapping significantly with other disease communities. These differences highlight that women tend to experience a more complex form of multimorbidity, whereas men's disease patterns are somewhat more compartmentalized.

Fig 4 shows a comparison of the prevalences for the six patterns detected for women and men. In general, the prevalence of multimorbidity patterns differs significantly between men and women (statistical significance was computed at the 95% confidence level). Only in cardiovascular (5.0% in both men and women) and respiratory diseases (5.9% in men; 7.3% in women) were no significant differences observed. However, men exhibit a higher prevalence of cardiometabolic diseases (43.4% in men; 31.0% in women), while women show a higher prevalence of mental (6.6% in men; 12.0% in

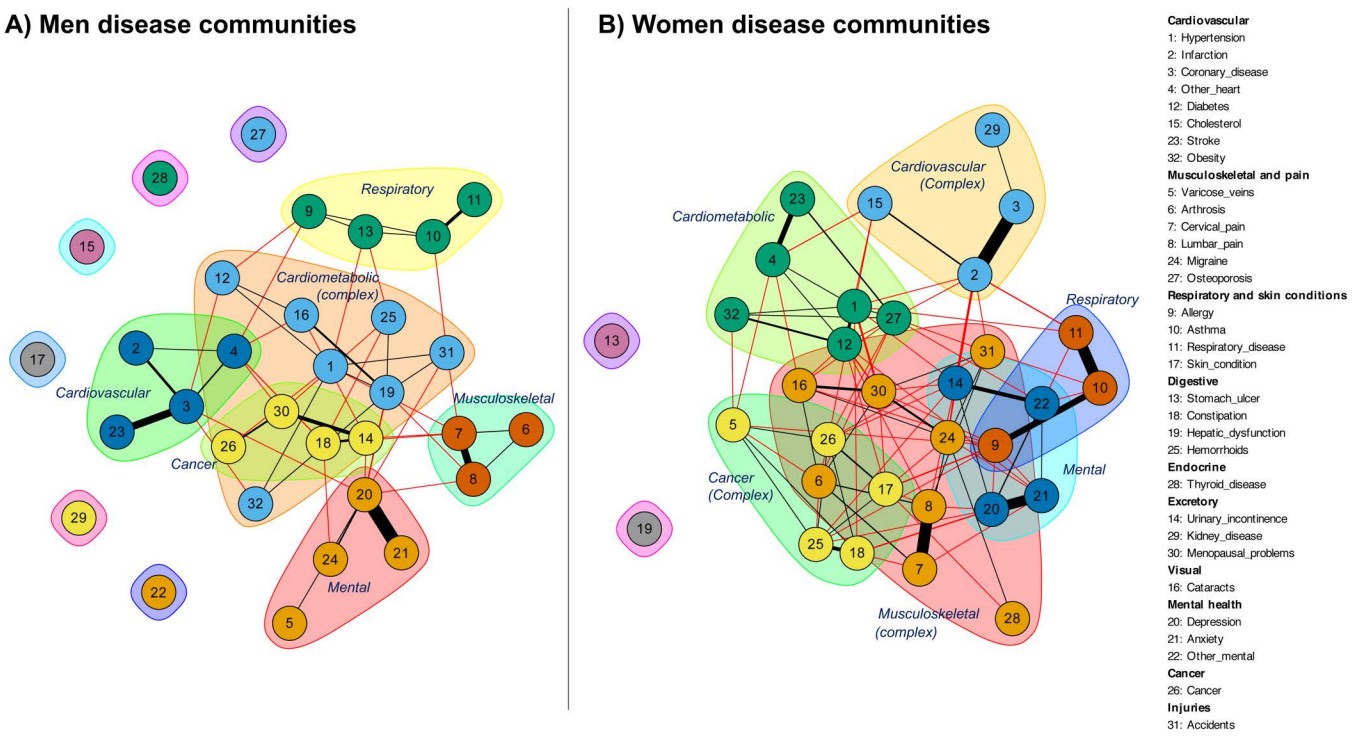

**Fig 3. Disease communities and betweenness analysis by gender.**

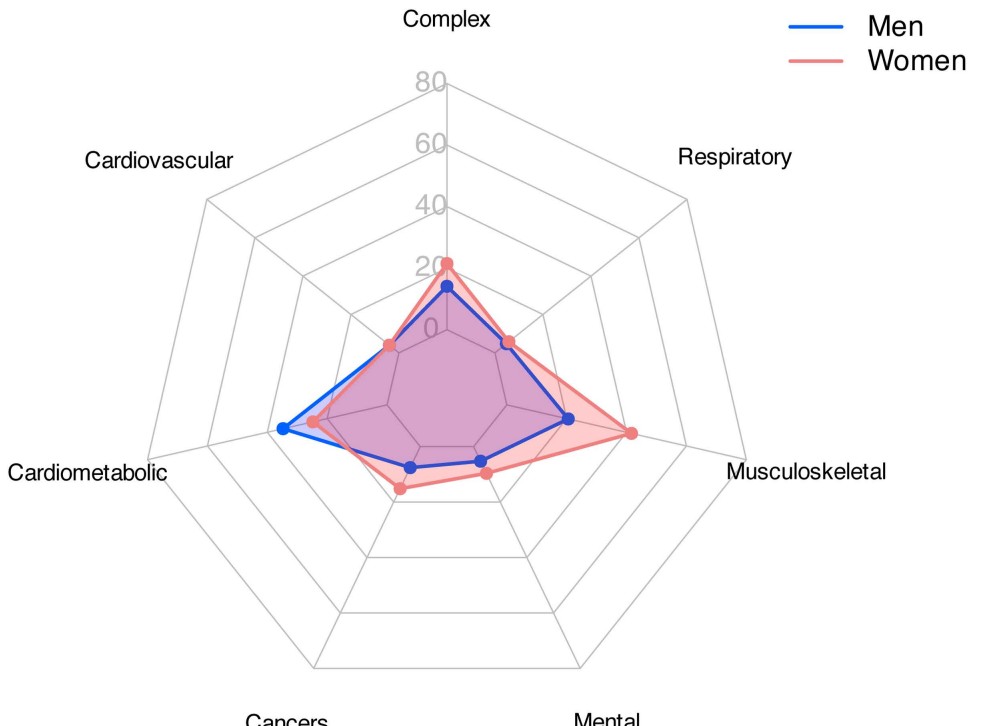

**Fig 4. Differences in the prevalence of the six identified multimorbidity patterns in men and women.**

women), cancer (9.5% in men; 19.0% in women), and musculoskeletal diseases (25.6% in men; 52.1% in women). These findings underscore again the importance of gender-based analyses to better understand and address the specific health needs of men and women.

From the combination of these multimorbidity patterns we obtained a seventh complex profile that was derived from their combination (i.e., this multisystem pattern resulted from the combination of two or more multimorbidity profiles between the six previously obtained). The differences in this pattern were also significant, with 17.6% of men classified in this multisystem pattern compared to the higher 26.9% of women. These findings further highlight both the greater burden of multimorbidity in women and the increased complexity of their chronic disease profiles.

### Identifying bridging diseases

To understand the relevance of the different diseases in the whole disease network we performed an exhaustive centrality analysis. Fig 5 illustrates centrality metrics for diseases within the men's network. Depression (D20) consistently shows high centrality across multiple measures. This condition presents the highest scores in degree, betweenness, PageRank, eigenvector centrality, and hub score. This indicates that depression is a crucial connector within the network, playing a significant role in bridging other conditions while being a central disease in its respective multimorbidity pattern. Hypertension (D1) also shows high betweenness and closeness centrality, signifying its importance in connecting various disease clusters and their diseases. Coronary disease (D3) and hepatic dysfunction (D19) exhibit high PageRank and overall centrality, indicating that they have numerous connections with other diseases that are also influential. The consistent presence of cardiovascular and metabolic conditions in top centrality rankings highlights their central role in the men's

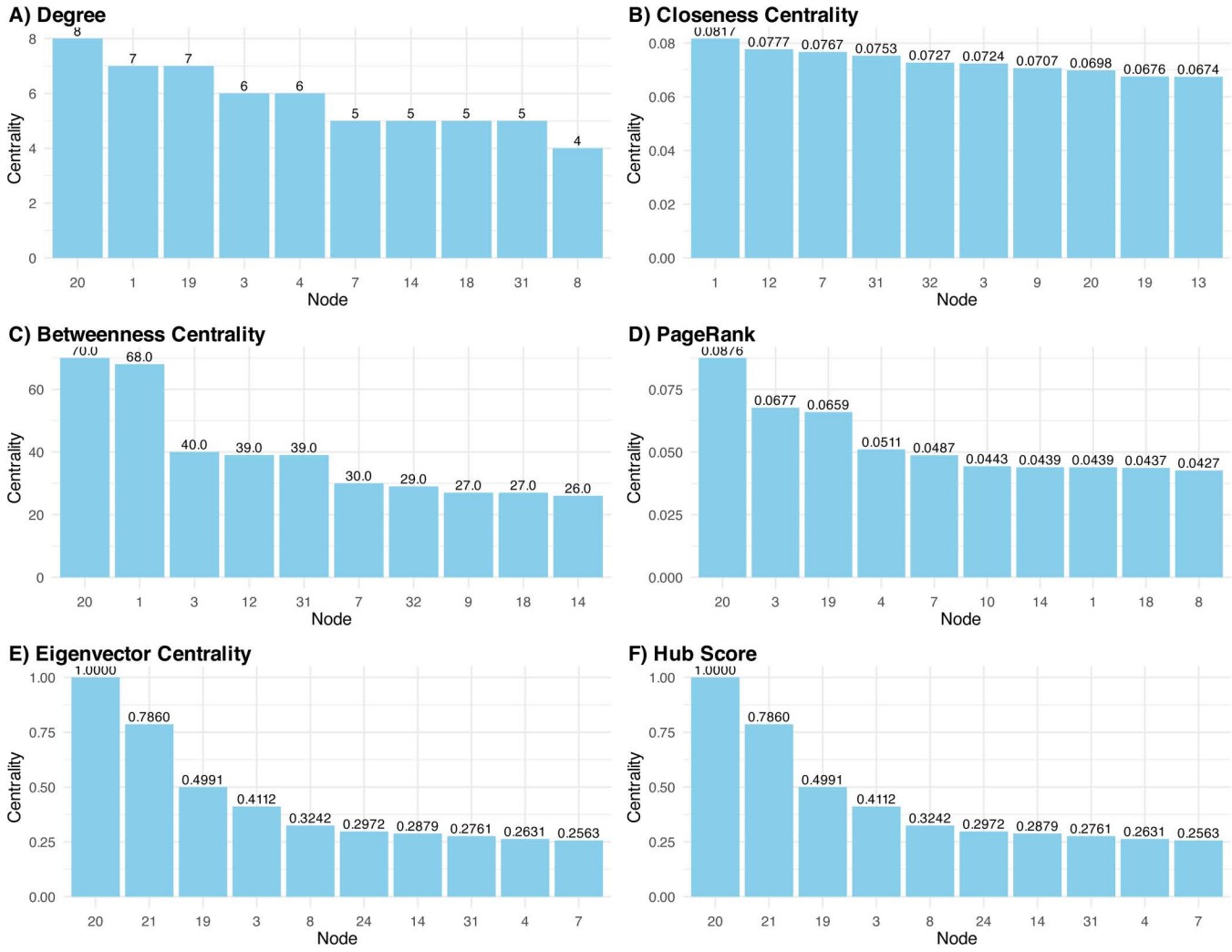

**Fig 5. Disease centrality analysis of men network.**

multimorbidity patterns, suggesting these conditions might be key targets for possible interventions to prevent and manage multimorbidity in this population group.

For women, Fig 6 presents a more complex multimorbidity landscape. The most connected diseases include hypertension (D1), diabetes (D12), and migraine (D24), all of which rank high in closeness centrality. Hypertension exhibits the highest betweenness centrality, making it the primary connector between disease communities and contributing to multisystem multimorbidity (or complex patterns). Depression (D20) also shows high eigenvector centrality, PageRank, and hub score, reaffirming its role as a key bridging disease in women.

### Association between bridging diseases and complex multimorbidity

Once the relevant variables were identified from the different centrality metrics, a logistic regression model was carried out to identify the effect of these variables in the configuration of multisystem patterns, that is, in patterns of complex multimorbidity that would combine two or more multimorbidity patterns (Table 1). The results of the analysis showed strong

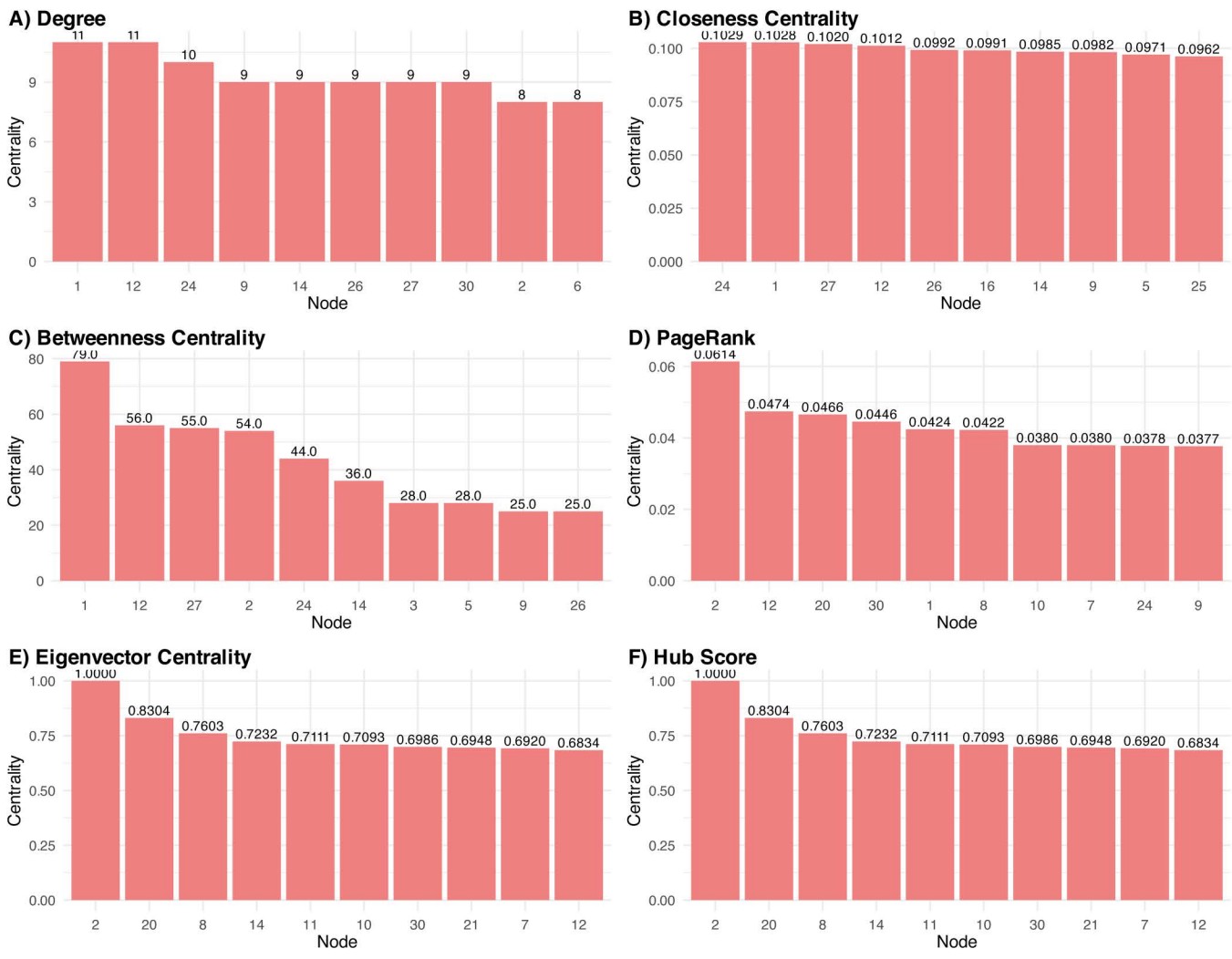

**Fig 6. Disease centrality analysis of women network.**

associations between several chronic conditions and the likelihood of presenting a multisystem multimorbidity pattern. For instance, cardiovascular conditions, particularly coronary disease (OR: 13.979 in men; 8.942 in women), and cardiometabolic diseases like obesity (OR: 4.417 in men; 6.205 in women) and hypertension (OR: 5.342 in men, 6.615 in women) revealed statistically significant contributions. Musculoskeletal disorders, such as cervical pain (OR: 12.235 in men; 5.235 in women), and lumbar pain (OR: 9.739 in men; 3.411 in women), were also highly associated with the Complex multimorbidity pattern. Mental health conditions like depression and anxiety were notable contributors, with depression being particularly influential in women (OR: 7.812) and also significant in men (OR: 3.073). These differences suggest that in women, mental health conditions are more central to the formation of complex disease patterns, while in men, their influence, although present, is less determinant.

Hepatic disfunction is the indicator that presents the strongest association with Complex multimorbidity in men (OR: 43.538) which highlights the overall relevance of this particular disease in the men group. On the other hand, osteoporosis is strongly associated with complex multimorbidity in women (OR: 8.928) but shows no significant association in men,

**Table 1. Logistic regression model to explain the complex (i.e., multisystem) multimorbidity pattern.**

| | | Men | | Women | |
|---|---|---|---|---|---|
| | **Complex** | **OR** | **[95% CI]** | **OR** | **[95% CI]** |
| Cardiovascular | Infarction | 4.578*** | (1.778 - 11.787) | 11.093** | (1.642 - 74.948) |
| | Coronary disease | 13.979*** | (4.148 - 47.113) | 8.942*** | (2.483 - 32.202) |
| | Other heart | 6.208*** | (2.445 - 15.761) | 4.925*** | (2.418 - 10.032) |
| Cardiometabolic | Obesity | 4.417*** | (2.197 - 8.881) | 6.205*** | (3.627 - 10.617) |
| | Hypertension | 5.342*** | (2.523 - 11.311) | 6.615*** | (3.809 - 11.488) |
| | Diabetes | 4.145*** | (1.978 - 8.685) | 7.322*** | (3.897 - 13.759) |
| Musculoskeletal | Cervical pain | 12.235*** | (5.341 - 28.027) | 5.235*** | (2.932 - 9.348) |
| | Lumbar pain | 9.739*** | (4.415 - 21.480) | 3.411*** | (1.952 - 5.960) |
| | Migraine | 3.804** | (1.136 - 12.741) | 3.101*** | (1.642 - 5.854) |
| | Osteoporosis | 1.229 | (0.034 - 45.052) | 8.928*** | (4.478 - 17.797) |
| | Accidents | 5.810*** | (2.171 - 15.547) | 1.561 | (0.563 - 4.329) |
| Mental | Depression | 3.073** | (1.023 - 9.228) | 7.812*** | (4.089 - 14.925) |
| | Anxiety | 3.725** | (1.101 - 12.601) | 5.879*** | (3.170 - 10.901) |
| Excretory | Urinary incontinence | 1.963 | (0.787 - 4.895) | 2.946*** | (1.380 - 6.292) |
| Digestive | Hepatic dysfunction | 43.538*** | (6.056 - 313.030) | 0.489 | (0.074 - 3.216) |
| Age (Ref. 50–64) | 65-74 | 3.910*** | (1.598 - 9.565) | 1.547 | (0.792 - 3.019) |
| | 75+ | 2.898*** | (1.154 - 7.278) | 3.172*** | (1.624 - 6.195) |
| Income (Ref. Q1) | Q2 | 0.968 | (0.409 - 2.292) | 1.102 | (0.598 - 2.032) |
| | Q3 | 0.683 | (0.274 - 1.699) | 1.253 | (0.611 - 2.572) |
| | Q4 | 0.617 | (0.220 - 1.726) | 0.756 | (0.293 - 1.950) |
| Education (Ref. Primary) | Secondary | 1.047 | (0.487 - 2.252) | 2.182*** | (1.204 - 3.957) |
| | Tertiary | 0.385 | (0.119 - 1.245) | 2.375*** | (1.034 - 5.455) |
| | Const. | 0.000*** | (0.000 - 0.002) | 0.001*** | (0.000 - 0.002) |

Note: statistically significant Odd Ratios at the level of 0.001, 0.01, and 0.05 are defined by ***, **, and *, respectively.

likely reflecting the higher prevalence and impact of this chronic condition in postmenopausal women. In contrast, accidents are a significant determinant of Complex multimorbidity for men (OR: 5.810), but not for women, possibly indicating a gendered difference in risk behaviors linked to physical injuries or trauma.

Age had a positive relationship with the complex pattern, i.e., the older the age, the greater the probability of having a complex pattern in both men and women. No statistically significant differences were found based on income, but statistically significant differences were found for education in the group of women.

## Discussion

### Summary of key findings

The present study highlights the relevance of network approaches for understanding the complexity of multimorbidity patterns in men and women. While predictive models of multimorbidity patterns may be valid for classifying concurrent disease clusters [10,18], network models based on LASSO regressions with penalties such as MGMs offer the possibility of capturing clinically meaningful multimorbidity patterns while eliminating spurious relationships that are usually included in conventional correlational analyses [21]. In this way, the networks obtained are more manageable and easier to interpret without losing sight of the internal structure of relationships (as is usually the case in latent class models or other clustering techniques commonly used in this field of knowledge). Moreover, the high accuracy of the MGM reflects their inherent

robustness, suggesting that they are effective in capturing the complex relationships between chronic diseases. Additionally, unlike other studies in the field of multimorbidity, this work focuses on the identification of what we have called "bridging diseases" (i.e., multi-connector diseases that serve as a key between the different patterns). This approach, from a health management perspective, offers the advantage of identifying possible intervention points to disrupt pathways leading to complex chronicity. In short, a different approach to that which has been carried out in recent studies of patterns of multimorbidity and which can help us in the planning of strategies to combat different forms of multipathology and, particularly, for complex forms of chronicity those that can be more harmful to people's health [6,17,26].

## Comparison with previous studies

Unlike other similar studies that incorporate MGM models for the study of multimorbidity patterns [21], our work goes a step further by incorporating additional metrics such as PageRank, Eigenvector centrality and Hub score that provide us with more comprehensive information on the relevance of different chronic diseases. We also incorporate precise measures of the effects of those conditions that we have called bridging diseases on the subsequent configuration of multisystem patterns, which have been recently identified as those that present a stronger relationship with mortality and health service frequentation [4].

Our analysis reveals striking differences in the prevalence of health conditions between men and women aged 50 and older. These initial results point to a higher disease burden in women, with a higher overall prevalence of chronic diseases, highlighting important gender inequalities in health [27,28] that subsequently translate into the resulting multimorbidity patterns obtained from the disease community analysis. Thus, the lower disease burden in the men's group makes it possible to obtain a more dispersed disease network, while on the contrary, the visualization and interpretation of the connections becomes more challenging in the women's group, who in general present more complex disease combinations (e.g., cardiovascular, cancer and musculoskeletal). Likewise, the overlap between the different multimorbidity patterns is much greater than in women, making the interpretation of the relationships between diseases and their communities more difficult. A greater complexity in the chronicity profiles of women is not as evident in studies using clustering models or latent classes to classify multimorbidity [4,6,10].

On the other hand, our exhaustive analysis of centrality metrics has allowed us to identify the diseases that may play a relevant role in the conformation of the main multimorbidity patterns (i.e., cardiovascular, cardiometabolic, musculoskeletal, mental, respiratory, and cancer). At the same time, we have examined the impact of high-centrality diseases in complex multimorbidity forms (i.e., in the multisystem pattern that is formed by two or more conditions). According to our findings, highly prevalent and preventable diseases such as obesity, hypertension, diabetes, anxiety, or depression play a fundamental role in the configuration of complex patterns of multimorbidity both in men and women. However, the clear example of relevant health conditions in the configuration of complex multimorbidity patterns is that of hepatic dysfunction, cervical/lumbar pain, and accidents in men, and osteoporosis, diabetes, urinary incontinence, and mental health-related conditions in women [29], which are partially linked to existing gender inequalities. For instance, the prominent role of liver disease in men's health outcomes in southern Spain has been observed in previous studies [4]. However, our study goes a step further by identifying this condition–preventable in many of the cases linked to alcoholism problems–as a trigger that could determine patterns of complex multimorbidity.

## Clinical and public health implications

These differences in the outcomes of multimorbidity patterns of men and women underscore the importance of gender-specific approaches in healthcare planning and management, ensuring that both physical and mental conditions are addressed according to predominant patterns in each sex while also considering the intermediary role certain diseases may play in the future development of multimorbidity patterns. [21]. Our results show that women tend to experience a more complex form of multimorbidity linked to cardiometabolic and mental profiles (even among high educated

women), whereas men's disease patterns are somewhat more compartmentalized and defined, commonly through cardiovascular and musculoskeletal patterns. These findings underscore the need for tailored healthcare interventions that address the distinct multimorbidity patterns in men and women [30]. By combining the identification of visual multimorbidity patterns with bridging diseases, this approach could detect potential disease targets, facilitating the implementation of gender-sensitive health strategies adapted to different multipathology profiles.

In any case, it should be noted that the disparities in the prevalence of diseases in these groups may also be related to differences in the use of services by women and men, and consequently to the subsequent obtaining of differential diagnoses in the health centers [31]. In fact, this is an issue that should be studied in depth in future qualitative studies incorporating additional information that could contribute to a better adjustment of the relational and classification models. Similarly, our work highlights the need to address the temporal dimension in the development of joint chronic diseases, a longitudinal perspective that could provide us with clearer evidence on the pathways to the various patterns of multimorbidity.

### Limitations and future direction

Although the results provide us with a different approach to multimorbidity, our work has a fundamental limitation associated with the cross-sectional nature of our study, which does not allow us to draw conclusions about the sequence of diseases that lead to the composition of the different patterns. Future research should employ longitudinal designs to trace multimorbidity progression over time, enabling a deeper understanding of the role of bridging diseases in shaping chronic disease pathways. Specifically, it would be essential to trace the disease trajectory that determines the final disease network. In any case, our work represents a first step in the identification of bridging diseases that, despite being preventable, enable the overlapping of complex patterns and whose future monitorization would make possible a better prognosis in chronic patients.

### Conclusion

This study provides valuable insights into the complexity of multimorbidity patterns in men and women, emphasizing the usefulness of network approaches in capturing these relationships. By utilizing network science methods, we have provided a clear characterization of the chronicity structures of these population groups while identifying bridging diseases that play crucial roles in linking different disease patterns and could serve as targets for clinical interventions aimed at disrupting the progression toward more severe multimorbid conditions. Our work also highlights gender-specific inequalities in multimorbidity patterns, with women experiencing a higher burden of chronic diseases, leading to more complex and overlapping disease networks. These findings underscore the importance of gender-specific health care strategies that address the predominant disease patterns in each sex while considering the intermediary roles of bridging diseases may play in the evolution of multipathology in older populations.

### Author contributions

**Conceptualization:** Javier Alvarez-Galvez.

**Data curation:** Javier Alvarez-Galvez.

**Formal analysis:** Javier Alvarez-Galvez, Javier Arroyo.

**Investigation:** Javier Alvarez-Galvez, Javier Arroyo.

**Methodology:** Javier Alvarez-Galvez.

**Project administration:** Javier Alvarez-Galvez.

**Supervision:** Javier Arroyo.

**Validation:** Javier Alvarez-Galvez.

**Visualization:** Javier Alvarez-Galvez.

**Writing – original draft:** Javier Alvarez-Galvez.

**Writing – review & editing:** Javier Arroyo.

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
