## [Decision Letter · Decision Letter 0]

14 Jan 2025

PONE-D-24-50745Uncovering Bridging Diseases in Complex Multimorbidity Pathways: A Network Science ApproachPLOS ONE

Dear Dr. Alvarez-Galvez,

Thank you for submitting your manuscript to PLOS ONE. After careful consideration, we feel that it has merit but does not fully meet PLOS ONE’s publication criteria as it currently stands. Therefore, we invite you to submit a revised version of the manuscript that addresses the points raised during the review process.

We look forward to receiving your revised manuscript.

Kind regards,

Kate E Dibble, Ph.D.

Academic Editor

PLOS ONE

Journal Requirements:

2. In the ethics statement in the Methods, you have specified that verbal consent was obtained. Please provide additional details regarding how this consent was documented and witnessed, and state whether this was approved by the IRB

4. In the online submission form, you indicated that the datasets generated and/or analysed during the current study are not publicly available due to the state of the project DEMMOCAD but are available from the corresponding author on reasonable request.

6. We note you have included a table to which you do not refer in the text of your manuscript. Please ensure that you refer to Table 1 in your text; if accepted, production will need this reference to link the reader to the Table.

Additional Editor Comments:

Good morning,

We recommended a major revision for this manuscript as there are a number of outstanding areas for improvement. Please see the reviewer comments for more suggestions/edits.

If you have any questions, please let me know.

Dr. Dibble

Reviewers' comments:

Reviewer's Responses to Questions

**Comments to the Author**

1. Is the manuscript technically sound, and do the data support the conclusions?

Reviewer #1: Yes

Reviewer #2: Yes

2. Has the statistical analysis been performed appropriately and rigorously? 

Reviewer #1: Yes

Reviewer #2: I Don't Know

3. Have the authors made all data underlying the findings in their manuscript fully available?

Reviewer #1: Yes

Reviewer #2: Yes

4. Is the manuscript presented in an intelligible fashion and written in standard English?

Reviewer #1: Yes

Reviewer #2: Yes

5. Review Comments to the Author

Reviewer #1: The manuscript provides valuable insights into multimorbidity patterns using network science methods, and the data appears to support the conclusions. However, I have a few suggestions:

Abstract: I suggest revising the introduction of the abstract to provide a clearer context and set the stage for the study’s objectives. The abstract currently jumps into the methodology without giving enough background on the significance of the study. A more comprehensive introduction will help the reader better understand the relevance of the research. Consider adding a sentence or two discussing multimorbidity profiles to provide context and highlight the study’s importance.

Presentation and Clarity: The manuscript is detailed but could benefit from minor rephrasing for clarity. A quick review for grammatical errors is recommended.

Gender-Specific Health Inequalities: The study's focus on gender differences is important. Expanding on how the findings can inform practical healthcare strategies for both sexes would be valuable.

Future Directions: A suggestion for future studies is a longitudinal approach to understand disease trajectories and their role in multimorbidity progression.

Reviewer #2: Review report

Title: “Uncovering Bridging Diseases in Complex Multimorbidity Pathways: A Network Science Approach”

Dear Editor, thank you for the opportunity to review this manuscript. The papers characterized patterns of multimorbidity and identified the bridging 20 diseases that facilitate the formation of complex multimorbidity patterns using network sciences. The paper addresses important public health aspects to understand complex multimorbidity patterns. The paper also readable. However, addressing the following points may benefit the manuscript.

General comments

• There are some clarity issues than need to be adressed. e.g., “...so the absence connections implies that there are no relationships or that they are too weak to be significant (line 140)”, “There indicators were binary and …(line 115) etc.

• Same issues are there, particularly in method sections.

• There are also long sentences that don’t allow to breath. e.g., line 122 to 1226

Methods

• Could you provide for using an estimation error of ±2.5 units when a 5% error margin is more commonly applied?

• After determining a minimum sample size of 2,200, why didn’t you apply your selection criteria (e.g., participants with two or more chronic conditions)?

• The calculation of the response rate is unclear. You planned to interview a minimum of 2,200 individuals, but ended up with 1,592 respondents with multimorbidity. What was the size included in your analysis ? or How does this translate to a response rate of only 13.3% (line 90)? Could you please described adequacy of your sample size, because of such analysis requires a bit largers sample size to reach cetrain conclusion ?

• The description of sample size, inclusion and exclusion criteria, and data collection procedures would benefit from being placed under separate subheadings rather than being mixed into the Study Design section.

• While you described how sparsity is achieved through the networking process, could you elaborate on how conditional independence assumptions were met? This is particularly important for chronic conditions that frequently co-occur (e.g., depression and anxiety), as their relationships may obscure the effects of other conditions.

• Could you provide more detail on how relationships among diseases or conditions are represented in the graph? For instance, how do you effectively capture associations and describe the parameterization process?

• In the final section of your Data Analysis (line 184), it would be helpful to describe the logistic regression approach in greater detail, especially since it may differ from approaches applied to other types of data.

Results

• For better clarity, the results could be organized into subheadings, such as prevalence estimates from the mixed graphical models for men and women, centrality analyses results, and findings from the logistic regression.

6. PLOS authors have the option to publish the peer review history of their article (what does this mean?). If published, this will include your full peer review and any attached files.

Reviewer #1: No

Reviewer #2: **Yes: **Yitagesu Habtu

---

## [Author Response · Author response to Decision Letter 1]

21 Feb 2025

Response to reviewer #1:

1. Abstract:

Comment (C): The introduction of the abstract should provide a clearer context and set the stage for the study’s objectives.

Response (R): We have revised the abstract to include a brief introduction on multimorbidity profiles and their significance in health research. This provides a better context before presenting the methodology and findings.

2. Presentation and Clarity:

C: Minor rephrasing for clarity and grammatical errors should be addressed.

R: We conducted a thorough proofreading of the manuscript, and we also have included subsections structure for better readability and clarity.

3. Gender-Specific Health Inequalities:

C: Expand on how findings can inform practical healthcare strategies for both sexes.

R: We have elaborated in the discussion section on how our findings can be used to inform gender-specific healthcare interventions.

4. Future Directions:

C: Consider suggesting a longitudinal approach to understand disease trajectories in multimorbidity progression.

R: We have added a section in the discussion suggesting future research directions, emphasizing the need for longitudinal studies to track multimorbidity evolution over time.

Response to reviewer #2:

1. Clarity Issues:

C: Some sentences need rewording for clarity.

R: As recommended, we revised sentences throughout the manuscript to improve clarity and conciseness.

2. Methods:

C: Justify the use of an estimation error of ±2.5 units when a 5% margin is more common.

R: Thanks for this observation. We have provided an explanation in the methods section for choosing ±2.5 units, emphasizing the need for higher precision in estimating multimorbidity patterns.

C: Explain why selection criteria were not applied after determining a minimum sample size of 2,200.

R: We clarified that the full sample was initially considered (to have a representative sample of people in this area, including also 50+ people without multimorbidity), and selection criteria were applied subsequently to focus exclusively on individuals with multimorbidity.

C: Clarify the calculation of response rate.

R: We have included a more detailed explanation of how the response rate was calculated, addressing the difference between the total surveyed individuals and those included in the final analysis.

C: Improve the organization of sample size, inclusion/exclusion criteria, and data collection procedures.

R: These components have been reorganized under a separate subheading to enhance the readability and logical flow of sample size calculation and selection criteria.

C: Elaborate on how conditional independence assumptions were met.

R: Thanks for this observation. We have explained in the methods section that “our implementation of mixed graphical models (MGM) incorporates LASSO regularization to enforce sparsity, thereby ensuring that only the strongest partial correlations remain in the network structure. This approach improves interpretability by reducing false associations and capturing the most meaningful disease relationships […] in terms of conditional independence, MGMs use the structure of a Markov random field to represent conditional relationships between variables. Formally, the model ensures that a variable is conditionally independent of the rest of the variables given its neighborhood in the network.”

Thus, our methodology not only ensures a more accurate representation of multimorbidity structures but also strengthens the validity of inferred relationships by addressing the issues of indirect associations and multicollinearity, which are common limitations in traditional regression-based approaches.

C: Provide more details on how disease relationships are represented in the graph.

R: We expanded the methods section to describe the parameterization process and how disease associations were captured using the mixed graphical model (MGM).

C: Describe the logistic regression approach in greater detail.

R: We added a more comprehensive description of our logistic regression approach.

3. Results:

C: Organize results into subheadings for better clarity.

R: We restructured the results section with subheadings: (a) Disease prevalence Estimates, (b) Detecting multimorbidity patterns, (c) Identifying bridging diseases, and (d) Association between bridging diseases and multimorbidity patterns, making it easier for readers to navigate the results. Subsections have also been incorporated into methods and discussion sections.

We sincerely appreciate the valuable feedback provided by the reviewers. These revisions have strengthened our manuscript, making it more precise, structured, and impactful. We hope the changes address all concerns and look forward to your further feedback.

Best regards,

The Co-authors

---

## [Editor Report · Decision Letter 1]

3 Apr 2025

Uncovering Bridging Diseases in Complex Multimorbidity Pathways: A Network Science Approach

PONE-D-24-50745R1

Dear Dr. Alvarez-Galvez,

We’re pleased to inform you that your manuscript has been judged scientifically suitable for publication and will be formally accepted for publication once it meets all outstanding technical requirements.

Kind regards,

Kate E Dibble, Ph.D.

Academic Editor

PLOS ONE
---

## [Editor Report · Acceptance letter]

PONE-D-24-50745R1

PLOS ONE

Dear Dr. Alvarez-Galvez,

I'm pleased to inform you that your manuscript has been deemed suitable for publication in PLOS ONE. Congratulations! Your manuscript is now being handed over to our production team.

Kind regards,

on behalf of

Dr. Kate E Dibble

Academic Editor

PLOS ONE